# Physical Activity Knowledge and Personal Habits with Recommendations for Patients: Self-Assessment by Primary Care Physicians

**DOI:** 10.3390/healthcare12111131

**Published:** 2024-05-31

**Authors:** Vilija Bitė Fominienė, Martirija Fominaitė, Saulė Sipavičienė

**Affiliations:** 1Department of Sport and Tourism Management, Lithuanian Sports University, Sporto Str. 6, LT-44221 Kaunas, Lithuania; 2Faculty of Medicine, Lithuanian University of Health Sciences, A. Mickeviciaus Str. 9, LT-44307 Kaunas, Lithuania; martirija.fominaite@stud.lsmu.lt; 3Department of Health Promotion and Rehabilitation, Lithuanian Sports University, Sporto Str. 6, LT-44221 Kaunas, Lithuania; saule.sipaviciene@lsu.lt

**Keywords:** primary care physicians, physical activity, recommendations for patients, competence

## Abstract

Primary care physicians (PCPs) should be active and reliable promoters of physical activity (PA), but there is no strong evidence that their knowledge and personal habits contribute to this. The aim of this study was to evaluate the frequency of PA recommendations provided by PCPs to patients in terms of their self-assessed PA knowledge and personal habits. This study used a cross-sectional design and data were collected through a self-reported online questionnaire. The study sample consisted of 202 PCPs from a large Lithuanian city, Kaunas, of which 122 were females (60.4%) and 80 were males (39.6%). The data were analyzed using SPSS version 29 (Statistical Package for the Social Sciences) for Windows. The findings show that the frequency of recommendations related to providing PA to patients is statistically significantly dependent on PCP health-friendly or partially favorable PA habits, their self-assessed level of knowledge about physical activity, and their self-assessed competence related to providing PA recommendations to patients, but this is not statistically dependent on objectively assessed level of knowledge related to PA.

## 1. Introduction

Physical activity (PA) is a preventive measure intended for the protection of public health [1]. This activity can help alleviate the symptoms of at least 26 conditions, such as depression, anxiety, stress, back pain and others [2]. On the other hand, physical inactivity has been identified as a significant risk factor for premature mortality [3] and is the main risk factor of 35 pathological conditions [4]. These facts make the promotion of PA an increasingly important public health priority. However, for influencing the PA of members of society, it is important to develop appropriate policies and offer effective interventions. And this is not easy to do, because PA “behavior” is multifactorial, influenced by various factors—both personal and social or environmental [5]. In these groups of factors, the search for PA promotion agents, tools and systems in health-care services, as a priority action, becomes especially relevant today, as it must respond to the WHO Global Action Plan for PA Promotion 2018–2030 [6]. In these searches, health-care professionals become particularly important agents, as they have excellent opportunities both to advise patients on health-promoting behaviors and to include PA in treatment pathways [7]. Primary care physicians (PCPs), who are trusted sources of health information [8] with close to 80% of the population visiting each year for all sorts of health-related issues, should be considered particularly important promoters of PA [9] that use PA as first-line therapeutic opinion [10]. However, despite strong evidence supporting the benefits of PA for the prevention and treatment of chronic diseases, it is still rarely recommended by PCPs [10,11]. The reasons for this are various, ranging from issues related to the competence of doctors and their habits, belief in a personal role, and setting an example for various organizational factors [9,10]. The latter is well reflected by the fact that national documents often lack PA promotion guidelines and a clearly defined action plan that PCPs should use. There is also a gap in the scientific literature, as there is still little analysis of the content of doctors’ recommendations on promoting PA to patients, the frequency of the recommendations themselves, and especially in the context of the doctors’ own behavior and health [8,12]. However, one of the most important predictors of PCPs providing health promotion counseling is PCPs’ own health and active behaviors, as it is difficult to encourage behaviors that one does not engage in [13], even though a considerable number of studies analyze the health-promoting PA or inactivity of various social groups in the context of their weight or BP [14,15,16]. Unfortunately, there is not much scientific research on PCPs’ BP or BMI and their PA [17,18,19,20,21,22,23], and even fewer studies have examined the relationship between PCPs’ PA and health indicators [24].

Importantly, there is a lack of scientifically based evidence demonstrating the relationship between PCPs’ PA and health indicators and recommendations made to patients [18,25,26], although some studies have indicated that in the presence of a lack of knowledge, which is a significant barrier to making recommendations regarding PA [9], PCPs’ greater level of PA and physical appearance qualify them as trustworthy advisors on PA matters [27,28,29]. In spite of this, some physicians doubt the value of counseling [30], and those who struggle with obesity or other health issues are reluctant to discuss PA with patients [29].

Regardless of the conclusions presented in the scientific literature, there is a lack of strong evidence to support the notion that PCPs’ behavioral habits in relation to health indicators and knowledge related to PA influence the frequency and nature of recommendations they make to patients. The aim of this study was to evaluate the frequency of PA recommendations provided by PCPs to patients in terms of their self-assessed PA knowledge and personal habits. In order to accomplish this aim, the subjects’ habits related to PA in terms of their weight and BP were analyzed. It was determined how the subjects self-assess their level of knowledge about PA and how they self-assess their competence in recommending PA to patients. Furthermore, the PCPs’ knowledge about PA was objectively evaluated and the frequency of actions during patient counseling on PA-related concerns was examined. It is likely that the results of this study will not only reveal the existing situation but will also become a prerequisite for developing innovative methods for increasing patients’ PA.

## 2. Materials and Methods

### 2.1. Research Design

A quantitative research strategy with a cross-sectional design was used to study the correlations between PCPs’ PA knowledge and personal habits and their PA recommendations to patients.

### 2.2. Participants

The sample for this cross-sectional study consisted of 202 PCPs working in Kaunas City Municipality. This choice was determined by the fact that the city is the second of the largest cities in Lithuania located in the central part of the country. It is home to the country’s largest university training health and life sciences specialists, and it operates one of the largest university hospitals in Lithuania. It is also the city where the number of practicing doctors per 10,000 patients is the highest in the whole country, amounting to 113.4 doctors per 10,000 patients (compared to 44.0 doctors per 10,000 patients in Lithuania as a whole). The number of PCPs is also the highest at 7.7 doctors per 10,000 patients [31]. The sample of the study was selected according to a simple random sampling method. The sample size was estimated by using Raosoft^®^ (Sample Size Calculator; Raosoft Inc., Seattle, WA, USA) [32], considering 5% as the margin of error and 95% as the confidence level, with a 50% response rate. The population considered was 424 PCPs. The inclusion criteria were the following: (1) working with adult patients with chronic diseases and (2) being a practicing physician.

### 2.3. Procedure

All data were collected via an online survey between October and December 2023. Potential participants were approached through personalized invitation using their work email. Prior to completing the survey, participants were introduced to the study aims and study measures and were told the approximate time needed to complete the survey (10–15 min). Those who agreed to participate were asked to complete the survey electronically (www.apklausa.lt), but before starting, they gave their informed consent for inclusion. All questions were set as mandatory, which prevented missing data. This study was conducted in accordance with the Declaration of Helsinki, and the protocol was approved by the Research Ethics Board of Lithuanian Sports University (protocol SMTEK-201, 10 November 2023).

### 2.4. Measures

The questionnaire consisted of 6 parts that assessed the sociodemographic characteristics of the PCPs (sex, age, and work experience); health indicators; PA habits; knowledge of and competence in PA; and frequency of actions when counseling patients on PA issues. Body mass index (BMI) and blood pressure (BP) were taken into consideration when assessing the PCPs’ health indicators.

BP was recorded using two values: systolic pressure (SBP) and diastolic pressure (DBP) in units of millimeters of mercury (mmHg) using the self-reported information provided by the participants. In accordance with the European Society of Hypertension (ESH)’s suggested classification [33], the participants were categorized as having optimal BP, normal BP, high-normal BP, grade 1 hypertension, grade 2 hypertension, or grade 3 hypertension.

BMI was calculated as body weight in kilograms (kg) divided by height in meters squared (m2) using self-reported information on body weight and height. In accordance with the WHO’s suggested classification, the respondents’ BMI was classified into the following categories: underweight (<18.5 kg/m^2^), normal weight (18.5–24.9 kg/m^2^), overweight (25.0–29.9 kg/m^2^), and obese (≥30.0 kg/m^2^) [34]. The Health and Exercise subscale of the Lifestyle and Habits Questionnaire—Brief was used to measure the participants’ health-promoting behaviors (LHQ-B) [35]. In this subscale, 6 items were rated on a five-point Likert scale ranging from 1 (never) to 5 (very often). In the assessment, according to the sex of the respondents, range categories (bottom, middle, and top) were distinguished. The higher the score, the more favorable a participant’s health-promoting behavior related to their PA habits. Cronbach’s alpha for the subscale in this study was 0.87.

Based on the WHO’s 2022 recommendations for PA [36] and a previous survey [37], three questions (How many minutes of moderate intensity aerobic activity should adults do every week? How long should moderate intensity aerobic activity be every day for health benefits? How many times a week should be devoted to moderate-to-high intensity strength training?) were created for the assessment of knowledge regarding PA, each with one correct answer. To analyze the results of the knowledge questions, each correct answer was scored 1 point. If the score was 0 or 1 point, the participants’ knowledge was assessed as weak; if the score was 2 points, the participants’ knowledge was assessed as average; and if the score was 3 points, it was assessed as good.

To determine the opinion of the subjects about their competence related to PA recommendations to patients, a single question based on previous surveys [37,38] was developed for this study. PCPs were asked to self-evaluate their competence (Please give your opinion about the competence you have related to PA recommendations to patients). The possible answers were “Sufficient” and “Not sufficient.” To determine the opinion of the subjects about their level of knowledge about PA, a single question based on previous surveys [38,39] was developed for this study. PCPs were asked to self-evaluate their level of knowledge (Please give your opinion about your knowledge about PA you have). The possible answers were “Sufficient” and “Not sufficient”.

A set of five items taken from previously validated surveys instruments that analyzed the promotion of PA in patients [39,40,41,42,43] was created to measure the frequency of actions when counseling patients on PA issues (I ask about the patient’s PA; I advise the patient to look into the health benefits of PA; I educate the patient about PA; When prescribing treatment, I include specific recommendations for PA; I am interested in the patient’s progress related to his/her PA). The items were rated on a five-point Likert scale ranging from 1 (never) to 5 (very often). Higher scores represented more frequent counseling. Cronbach’s alpha for the subscale in this study was 0.82.

### 2.5. Statistical Analysis

Statistical analysis was performed using the statistical data processing software package SPSS version 29 (Statistical Package for the Social Sciences) for Windows. The obtained data (derived from the sociodemographic data in the first section of the survey) were used to compute the arithmetic mean (M). The percentage expression of these values was then used to compare the participant groups with one another in frequency tables. The normal distribution of the data was evaluated using the Kolmogorov–Smirnov test. The chi-squared test and Fisher’s exact test were used to compare qualitative values, and the z-test was used to assess the equality of proportions. The Pearson correlation coefficient was used to calculate correlations between the study variables. A binary logistic regression analysis was performed to determine the factors predicting recommendations about PA to patients, and the results were described using cross-odds ratios (ORs) and with their 95% confidence intervals (CIs). A *p*-value of less than 0.05 was considered statistically significant, and the values of the correlation coefficients were evaluated as follows: 0.0–0.3—poor, 0.3–0.5—fair, 0.6–0.8—moderately strong, and 0.8–1.0—very strong [44].

## 3. Results

Among the participants, there were 122 females (60.4%) and 80 males (39.6%) in the two age groups, with 75 participants (20.3% females and 16.8% males, respectively) being 40 years of age or younger, and 127 participants (40.1% females and 22.8% males, respectively) being older than 40. Furthermore, 17 (8.4%) participants had less than two years of experience, 58 (28.7%) had two to ten years of experience, and the remaining (62.9%) had more than ten years of experience. The descriptive statistics are displayed in Table 1. Upon evaluating the health indicators of the participants, it was discovered that the BMIs of the entire sample varied between 17.7 and 38.9 (mean = 25.3, SD = 4.0) kg/m^2^.

The majority of the participants (53.0%) were either overweight (41.1%) or obese (11.9%), with 1.5% being underweight. There were statistically significant differences between sexs (*p* < 0.001). More women (60.7%) than men (22.5%) had a normal body weight, while a larger proportion of men were overweight (61.3%). There was a statistically significant difference (*p* < 0.05) between different age and BMI groups: older respondents who were over 40 were more likely to be obese than younger ones.

The participants’ self-reported blood pressure (BP) data showed that most respondents had optimal BP (36.1%) and normal BP (29.7%). Meanwhile, only 7.4% of the participants had hypertension. When comparing the variables, statistically significant differences (*p* < 0.001) were observed in terms of sex.

After evaluating the participants’ PA and health-promoting practices, it was discovered that the majority of the respondents (44.1%) fell in the Middle category. A similar number of participants were in the Bottom category (26.7%) and Top category (29.2%). There were statistically significant differences (*p* < 0.001) based on the respondents’ sex. More females were in the Middle category (51.6%) and more males were in the Bottom category (41.3%). When comparing the data in terms of age, no statistically significant differences were found, but older participants demonstrated better PA habits more often than younger participants (34.6% of older participants and 20.0% of younger participants).

When comparing the participants’ PA habits and BP and BMI categories (Table 2), it was found that the participants’ PA habits were statistically significantly dependent on BP (*p* < 0.001) and BMI (*p* = 0.009). While participants who were overweight or obese were mostly in the Bottom group of PA habits (70.4%), those with optimal and normal blood pressure were more likely to exhibit PA habits in the Middle and Top categories (77.5% and 71.2%, respectively).

When evaluating the participants’ self-reported level of knowledge about PA (Table 1), it was found that regardless of age and sex, the majority of participants believed that their level was sufficient (73.3%). Upon assessing the participants’ objective knowledge regarding PA, it was found that only 7.9% of the participants exhibited good knowledge. Meanwhile, the objectively assessed knowledge of 45.0% of participants was average, and the objective knowledge of the majority of participants (47.0%) was weak. Sex and age had no impact on these findings. When the participants’ self-reported and objective levels of knowledge were compared (Table 3), a significant difference (*p* = 0.017) was discovered. A larger proportion of participants whose knowledge of PA was categorized as weak assessed their own knowledge as being sufficient (52%).

The self-reported competence level, which was associated with providing PA recommendations to patients (Table 1), of most participants was assessed as not sufficient (72.8%) and did not differ based on sex (*p* = 0.053) or age (*p* = 0.626).

As shown in Table 4, the correlation between self-reported knowledge level about PA and self-reported competence level was positive (r = 0.294; *p* < 0.01). On the other hand, objective knowledge level was negatively associated with self-reported knowledge level about PA (r = −0.154; *p* < 0.05).

After evaluating the frequency of PA recommendations given by the participants to patients (Table 5), it was found that the participants most often asked patients about their PA (3.28 ± 1.203) and advised patients to look into the health benefits of PA (3.22 ± 1.161). This was, however, performed moderately/infrequently in the entire sample. The participants were quite rarely interested in their patients’ progress related to their PA (2.53 ± 1.32). The frequency of making PA recommendations to patients was not statistically dependent on the objective level of knowledge, but statistically depended on the respondents’ self-reported knowledge, self-reported competence, and self-reported PA behaviors.

A binary regression analysis was conducted (Table 6), which allows us to predict that recommendations related to providing PA to patients will be more frequent when there is a higher self-reported knowledge level, self-assessed competence in providing PA recommendations to patients, and better PA habits.

## 4. Discussion

This study primarily assessed PCPs’ habits related to PA. According to the WHO recommendations, a person is required to have 150 min of moderate-intensity aerobic PA, at least 75 min of high-intensity aerobic PA per week, or an equivalent combination of moderate- and vigorous-intensity activity throughout the week. Additionally, it is advised to perform muscle-strengthening exercises involving all major muscle groups on two or more days per week at a moderate-to-high intensity [36]. Across the world, it is estimated that more than 27% of adults’ PA does not meet the level recommended by the WHO [1], which, like any other everyday behavior, is influenced by habits [36].

The findings of this study also showed that the PCPs had inadequate or very low levels of PA, which suggests that PA is either very low or nonexistent among doctors. Specifically, this study found that the majority of the respondents (44.1%) fell in the Middle category, meaning that while they have some PA-related skills, their behavior only partially complies with the WHO recommendations. A significant number of participants—26.7%—were classified in the Bottom category, meaning that they lack the abilities necessary for physical exercise, do not adhere to guidelines for PA, and demonstrate a sedentary lifestyle that is harmful to health.

Similar findings have been reported in a study assessing young medical residents, which found that 71% of the participants engaged in physical exercise less frequently than recommended [19]. Other studies also reported that PCPs’ levels of PA were insufficient, although the percentages regarding PA levels varied, ranging from 45.6% [22], 60.5% [17], and 65% [45] to 78.9% [46], 82.5% [13], and 86% [18] among the surveyed physicians.

Meanwhile, only 29.2% (26% of men and 31% of women) of the participants in this study consciously understand the importance of PA, regularly exercise and/or play sports, and follow the WHO recommendations for PA. Similar patterns were observed in the results of previous studies that examined PA among physicians, indicating that 21.1% [46] or 35% [45] of the participants were sufficiently physically active. A higher number of physically active participants has also been reported, reaching 39.5% [17] or even 54.4% [22]. All of these data can be associated with the specifics of studies in which limited education on physical activity prevails, with the COVID-19 crisis [47] and with the lack of free time, which is often determined by long working hours [48].

In terms of age, the results indicated that doctors over 40 engaged in greater levels of PA (34.6% versus 20% among doctors younger than 40). Although there are studies that did not identify a statistically significant correlation between age and PA [17], other studies [49] also suggest that the older group of PCPs may be more active than the younger group, and this relates to the time that early-career physicians can devote to social and recreational activities [50]. It was also established that having small children can be considered a risk factor for physical inactivity [51].

PA can also be influenced by sex, although this finding is not unequivocal [52,53]. According to this study, a higher percentage of men than women lacked physical exercise abilities, meaning they engaged in no PA at all. However, other studies have shown conflicting results, showing that male doctors are 56.8% more likely to be moderately physically active, while female doctors are 54.6% more likely to be inactive [17].

Studies have shown a correlation between an individual’s BMI and BP and their PA [54]. More than half of the doctors in our sample were overweight or obese, with men being more likely to be overweight or obese regardless of age. In contrast, the majority of participants with a normal body weight (45.5%) were women. These results are somewhat better compared to the findings of a study in 2022 on the entire population in Lithuania, which found that 25.4% of adults in Lithuania were obese, and 58.7% were overweight [55].

Similar findings from other studies that examined doctors’ weight features pointed to a tendency for doctors to be overweight or obese. Up to 50% of doctors may be overweight, depending on the nation, culture, or area of expertise of the participants [27]. However, other studies reported a lower number of overweight doctors in the range of 30.4% [25], 36.5% [56], 38% [57], 42.3% [17], and 46.1% [58]. Obesity rates among physicians ranged from 11.8% to 16.1% [25,27] or even up to 31% [17]. Even among young individuals, 46.4% of medical residents were overweight or obese [19]. These patterns indicate that physicians, like other segments of the population, are more likely to have a higher-than-ideal BMI, which is associated with a higher risk of non-communicable diseases and may also be a contributing factor to mortality, as these illnesses accounted for 5 million deaths in 2019 [55].

According to previous studies, in addition to an individual’s PA level, weight is also often associated with an individual’s BP [59]. In this study, no significant relationship between subjects’ weight and hypertension was found, which may be related to the sufficiently small number of subjects who had hypertension, i.e., only one of the twenty-nine participants were found to have hypertension. Nevertheless, the fact that physicians, including PCPs, are more prone to cardiovascular diseases, which are often associated with stress and unhealthy lifestyle [60], both of which are associated with insufficient PA, should not be ignored [61].

PA is a complex behavior, practice of which and recommendations to others largely depend on available knowledge [62]. Our study’s findings demonstrated that PCPs were not aware of the most recent PA recommendations. Just 8% of the respondents showed an excellent level of knowledge when objective knowledge of PA was examined. Another study [27] reported an even lower number of physicians who were highly aware of current PA recommendations (5.4%). This knowledge is also insufficient among medical students, since only 46.8–52% of students [63,64] know about the WHO guidelines. Moreover, just 70.9% of medical students acknowledge the value of PA for treating diseases, despite the fact that 94.8% of them think that it is crucial for preventing sickness [63].

Our study found a significant difference when comparing PCPs’ objective level of knowledge with their self-reported level of knowledge. It is important to mention that sex and age were not significant influencing factors, but it should be noted that the majority of participants whose PA knowledge was objectively categorized as weak assessed their knowledge as sufficient. Only 27% of the participants reported having sufficient competence, indicating that they did not rate their own knowledge as high as their competence level, which was related to making PA recommendations to patients. There was a positive correlation between self-reported knowledge level of PA and self-rated competence level. These results align with the findings of a previous study that discovered that although 59% of doctors felt inadequately informed about current PA recommendations, 54% of doctors felt competent and confident in their capacity to counsel patients on how to be appropriately physically active [26]. In another study, even though doctors agreed that PA was important, they were hesitant to offer counseling since they did not know how beneficial it would be [29].

In this study, when we evaluated recommendations related to PA, we found that PCPs most often asked patients about their PA and advised patients to look into the health benefits of PA, yet they did not do this frequently enough. In addition, PCPs were relatively infrequently interested in patients’ progress in relation to their PA.

The frequency of making PA recommendations to patients was significantly related to PCPs’ PA habits, self-reported knowledge level, and self-reported competence level, but it was not significantly related to objective knowledge level. A self-reported sufficient level of competence was associated with providing PA recommendations to patients, self-sufficient level of knowledge, and being in the categories of High or Medium PA habits. Therefore, sufficient competence level has a significant influence on the frequency of making PA recommendations to patients. While some studies [26,65] found no statistically significant difference between PCPs’ PA levels and their PA counseling practices, the majority of studies reported correlations between PCPs’ PA levels and frequency of consultations [48,66,67,68,69,70]. Patients are more likely to receive advice about the advantages of PA from doctors who have a track record of engaging in health-promoting behaviors, feel that they are competent to advise patients on sports or exercise, and have a solid understanding of PA. Objective knowledge of PA is not important, though. Compared to PCPs who are not physically active, those who are physically active are more likely to encourage their patients to be physically active [69,71]. Physicians who are physically fitter are better at persuading patients of the advantages of PA [72]. PCPs who engage in resistance exercise are five times more likely to discuss the advantages of this type of exercise with their patients. Similarly, PCPs who engage in aerobic exercise are nearly six times more likely to suggest it. Physicians who are not physically active, on the other hand, are less likely to be aware of the WHO recommendations for PA [21].

PCPs who practice healthy lifestyle behaviors could act as role models for patients and therefore provide more effective healthy lifestyle counseling [11,26,70]. When PCPs engage in PA, patients are more likely to accept their advice because they view them as more credible and inspiring role models [66]. Moreover, a more active preventive approach to PA can reduce the occurrence of chronic diseases by increasing PA among patients who are considered inactive [7].

Limitations: Our study had several limitations. First, data collection was carried out using a remote survey, so the response rate could not be accurately calculated. A self-reported questionnaire was used to gather the data, and such a method is sensitive to the possibility of bias in answers, particularly when evaluating one’s own health-promoting behavior and self-assessing one’s own knowledge and competence. Also, for PA assessment, we used only habit assessment, and the level of intensity of exercise was not gathered from the PCPs. Conclusions for the entire population cannot be drawn from this study because it was limited to one large city, which has the highest ratio of doctors to patients (10,000:1) and is home to the nation’s largest university clinical hospital as well as the largest university.

## 5. Conclusions

The results of this study revealed that the majority of participants exhibited poor or insufficient PA-related behaviors, which can be associated with elevated BP, and such behaviors may also be seen as a result of being overweight or obese. Despite the manifestations of unhealthy behavior related to PA and the observed increase in weight and BP revealed in the study, most of the participants stated that the knowledge level they personally have about PA is sufficient. Unfortunately, it was found that most of these physicians lacked objective knowledge about PA, and nearly three-quarters of participants felt that their experience was insufficient to recommend PA to patients. Also, the study revealed that the frequency of recommendations related to the provision of PA to patients is statistically significantly dependent on PCPs’ health-friendly or partially favorable PA habits, their self-assessed level of knowledge about PA, and self-assessed competence related to the provision of PA recommendations to patients, but this is not statistically dependent on objectively assessed level of knowledge related to PA. These results are thought to highlight the need for policies and initiatives that would enable physicians to continuously broaden their understanding of PA-related concerns and increase their familiarity with the potential integration of PA into treatment pathways. It would also be good to offer ways to assist in changing the way PCPs themselves behave when it comes to their health. In the long term, this is crucial for a healthier society.

## Figures and Tables

**Table 1 healthcare-12-01131-t001:** Sample characteristics by sex and age (*n* = 202).

	All *n* = 202	Sex			Age		
		Male *n* = 80 (39.6%)	Female *n* = 122 (60.4%)	*p*-Value	≤40 *n* = 75 (37.1%)	>40 *n* = 127 (62.9%)	*p*-Value
**Mean Height (cm)**	174.98	184.11	168.99	<0.001	176.19	174.27	0.176
*Std. deviation*	9.73	7.55	5.36		9.31	9.94	
**Mean Weight (kg)**	77.98	91.75	68.94	<0.001	75.80	79.26	0.143
*Std. deviation*	16.22	11.42	12.01		14.92	16.87	
**BMI (kg/m^2^)**	25.31	27.08	24.15	<0.001	24.29	25.91	0.006
*Std. deviation*	4.03	3.23	4.08		3.73	4.09	
**Underweight *n* (%)**	3 (1.5)	0 (0.0)	3 (2.5)	<0.001	3 (4.0)	0 (0.0)	0.047
**Normal weight *n* (%)**	92 (45.5)	18 (22.5)	74 (60.7)		36 (48.0)	56 (44.1)	
**Overweight *n* (%)**	83 (41.1)	49 (61.3)	34 (27.9)		31 (41.3)	52 (40.9)	
**Obese *n* (%)**	24 (11.9)	13 (16.3)	11 (9.0)		5 (6.7)	19 (15.0)	
**BP *n* (%)**				<0.001			0.24
**Optimal BP *n* (%)**	73 (36.1)	1 (1.3)	72 (59.0)		30 (40.0)	43 (33.9)	
**Normal BP *n*** **(%)**	60 (29.7)	34 (42.5)	26 (21.3)		22 (29.3)	38 (29.9)	
**High-normal *n* (%)**	54 (26.7)	38 (47.5)	16 (13.1)		21 (28.0)	33 (26.0)	
**Hypertension *n* (%)**	15 (7.4)	7 (8.8)	8 (6.6)		2 (2.7)	13 (10.2)	
**Category of PA habits *n* (%)**				<0.001			0.086
Bottom	54 (26.7)	33 (41.3)	21 (17.2)		23 (30.7)	31 (24.4)	
Middle	89 (44.1)	26 (29.2)	63 (51.6)		37 (49.3)	52 (40.9)	
Top	59 (29.2)	21 (26.3)	38 (31.1)		15 (20.0)	44 (34.6)	
**Objective level of knowledge *n* (%)**				0.773			0.279
Good	16 (7.9)	5 (6.3)	11 (9.0)		8 (10.7)	8 (6.3)	
Average	91 (45.0)	37 (46.3)	54 (44.3)		29 (38.7)	62 (48.8)	
Weak	95 (47.0)	38 (47.5)	57 (46.7)		38 (50.7)	57 (44.9)	
**Self-reported level of knowledge *n* (%)**				0.271			0.194
Sufficient	148 (73.3)	62 (77.5)	86 (70.5)		51 (68.0)	97 (76.4)	
Not sufficient	54 (26.7)	18 (22.5)	36 (29.5)		24 (32.0)	30 (23.6)	
**Self-reported competence, associated with providing PA recommendations to patients *n* (%)**				0.053			0.626
Sufficient	55 (27.2)	28 (35.0)	27 (22.1)		22 (29.3)	33 (26.0)	
Not sufficient	142 (72.8)	52 (65.0)	95 (77.9)		53 (70.7)	94 (74.0)	

**Table 2 healthcare-12-01131-t002:** Comparison of categories of PA habits and BP and BMI categories.

		BP			*p*-Value	BMI			*p*-Value
		Optimal and Normal	High Normal	Hypertension		Under-Weight	Normal	Overweight/ Obese	
Categories of PA habits	Bottom	22 (40.7)	24 (44.4)	8 (14.8)	<0.001	1 (1.9)	15 (27.8)	38 (70.4)	0.009
	Middle	69 (77.5)	15 (16.9)	5 (5.6)		1 (1.1)	43 (48.3)	45 (50.6)	
	Top	42 (71.2)	15 (25.4)	2 (3.4)		1 (1.7)	34 (57.6)	24 (40.7)	

**Table 3 healthcare-12-01131-t003:** Comparison of objective level of knowledge and self-reported level of knowledge.

		Self-Reported Level of Knowledge *n* (%)		*p*-Value
		Not Sufficient	Sufficient	
Objective level of knowledge *n* (%)	Weak	18 (33.3)	77 (52.0)	0.017
	Average	28 (51.9)	63 (42.6)	
	Good	8 (14.8)	8 (5.4)	

**Table 4 healthcare-12-01131-t004:** Correlations between variables.

Variables	SR Competence	SR Knowledge	O Knowledge
SR Competence	1		
SR Knowledge	0.294 **	1	
O Knowledge	−0.015	−0.154 *	1

Note: SR—self-reported; O—objective. * Correlation is significant at the 0.05 level (2-tailed). ** Correlation is significant at the 0.01 level (2-tailed).

**Table 5 healthcare-12-01131-t005:** Comparison of providing PA recommendations frequency to patients and categories of PA habits, knowledge about PA levels and levels of competency.

Providing PA Recommendations to Patients	TOTAL (M, SD)	Category of PA Habits			*p*-Value	Objective Level of Knowledge			*p*-Value	Self-Reported Level of Knowledge		*p*-Value	Self-Reported Competence		*p*-Value
		Bottom	Middle	Top		Weak	Average	Good		Not Sufficient	Sufficient		Not Sufficient	Sufficient	
Ask about the patient’s PA	3.28 1.20	2.80 1.21	3.20 1.15	3.85 1.05	<0.001	3.26 1.29	3.31 1.12	3.25 1.18	0.963	2.83 1.08	3.45 1.21	0.001	3.13 1.20	3.69 1.12	0.003
Advise the patient to look into the health benefits of PA	3.22 1.16	2.85 1.16	3.03 1.10	3.85 1.01	<0.001	3.24 1.24	3.21 1.12	3.19 1.18	0.973	2.91 1.07	3.34 1.18	0.019	2.99 1.16	3.84 0.92	<0.001
Educate the patient about PA	3.00 1.08	2.59 1.02	2.91 0.97	3.51 1.12	<0.001	3.05 1.15	2.96 1.03	2.94 0.99	0.809	2.56 0.95	3.16 1.09	<0.001	2.78 1.06	3.60 0.92	<0.001
When prescribing treatment, include specific recommendations for PA	2.79 1.20	2.56 1.24	2.55 0.99	3.37 1.29	<0.001	2.88 1.32	2.74 1.07	2.56 1.21	0.516	2.50 1.06	2.90 1.24	0.037	2.57 1.16	3.38 1.19	<0.001
To be interested in the patient’s progress related to his/her PA	2.53 1.32	2.22 1.31	2.17 1.03	3.36 1.37	<0.001	2.49 1.40	2.54 1.30	2.69 0.95	0.862	2.09 1.15	2.69 1.34	0.004	2.20 1.19	3.40 1.26	<0.001

Note: M—mean, SD—standard deviation.

**Table 6 healthcare-12-01131-t006:** Regression analysis of factors affecting provided PA recommendations to patients frequency.

Variables	Provided PA Recommendations to Patients Frequency		
	OR	95% CI	*p*-Value
Self-reported level of knowledge (ref = “weak/average”)	2.744	1.204–6.252	0.016
Objective level of knowledge (ref = “weak/average”)	1.665	0.543–5.102	0.372
Sex (ref = “women”)	1.041	0.535–2.028	0.906
Age (ref = “≤40”)	1.341	0.695–2.586	0.382
Category of PA habits (ref = “bottom”)	2.153	1.001–4.634	0.050
Self-reported competence, associated with providing PA recommendations to patients (ref = “not sufficient”)	4.076	1.992–8.340	<0.001

Note: OR—odd ratios, CI—confidence interval.

## Data Availability

The data that support the results are available from the corresponding author upon reasonable request.

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
