# Peer review of "Physical Activity Knowledge and Personal Habits with Recommendations for Patients: Self-Assessment by Primary Care Physicians"

_healthcare, 2024, doi:10.3390/healthcare12111131_

Round 1
Reviewer 1 Report
Comments and Suggestions for Authors
The paper contains fundamental distortions concerning the essence of the phenomena studied. In general, it should be stated that in a study concerning, for example, knowledge or personal habits, we cannot treat the obtained results as facts, but only as declarations of our own opinions (self-assessments). This reservation should be immediately indicated in the title of the paper, otherwise it must be assumed that these features were actually checked by the use of specialized tests verifying knowledge.
Specific comments:
1. Correlations, mentioned as statistical tools, are not a guarantee of cause-and-effect relationships.
2. Why were the tests carried out and special attention paid to the pressure test? From the great repertoire of diagnostic tests on the basis of which the state of health is diagnosed, dozens of others can be selected. Why such a special preference for pressure measurement?
3. Asking doctors 3 questions in a gradation of 1,2,3 points, regarding the self-assessment of their knowledge is not diagnostic, and even inclines to mislead. Many doctors are probably ashamed of their ignorance and lack of personal habits and precaution in marking 2. This is also the level at which patients are advised to behave in this way.
4. There is frequent repetition of data exchanged in the text and in the tables.
5. Conclusions are not conclusions, but only a summary of the results and data from the discussions. There is no indication of actions that could be taken on the basis of the obtained, albeit doubtful, facts.
Author Response
Dear Reviewer,
Thank you very much for taking the time to review this manuscript. You will find the answers below, and the relevant changes/corrections are included in the resubmitted version of the article.
The paper contains fundamental distortions concerning the essence of the phenomena studied. In general, it should be stated that in a study concerning, for example, knowledge or personal habits, we cannot treat the obtained results as facts, but only as declarations of our own opinions (self-assessments). This reservation should be immediately indicated in the title of the paper, otherwise it must be assumed that these features were actually checked by the use of specialized tests verifying knowledge
Thanks for this note. We took it into account and accordingly adjusted both the title, the aim of the study, and the conclusions.
Correlations, mentioned as statistical tools, are not a guarantee of cause-and-effect relationships.
In this study, we analyzed correlations between SR Competence, SR Knowledge and O Knowledge and we only indicated that there is a a relationship or association between two variables. We have not explored or confirmed the causality, because correlation does not imply causation. Further investigation is needed in order to guarantee cause-and-effect relationships, we have only identifyed potential relationships.
Why were the tests carried out and special attention paid to the pressure test? From the great repertoire of diagnostic tests on the basis of which the state of health is diagnosed, dozens of others can be selected. Why such a special preference for pressure measurement?
When evaluating the subjects' PA, we analyzed it in terms of a person's weight and BP. This is carried out in a considerable part of the studies analyzing the PA of a person. Added in text See lines 59-60.
Asking doctors 3 questions in a gradation of 1,2,3 points, regarding the self-assessment of their knowledge is not diagnostic, and even inclines to mislead. Many doctors are probably ashamed of their ignorance and lack of personal habits and precaution in marking 2. This is also the level at which patients are advised to behave in this way.
Using the 3 questions you mentioned about PA, we objectively determined the knowledge of PCPs, where there was only one correct answer - there is no gradation and there were 4 possible answers. We have chosen 3 main test questions prepared according to WHO recommendations. And the self-assessment took place not by using the grading system, but by indicating whether their knowledge is sufficient or not sufficient – they had to choose between these answers. Therefore, we believe that this test accurately helped us diagnose they objective knowledge.
There is frequent repetition of data exchanged in the text and in the tables.
Errors crept into the text and tables, indicating the wrong data. They are fixed.
Conclusions are not conclusions, but only a summary of the results and data from the discussions. There is no indication of actions that could be taken on the basis of the obtained, albeit doubtful, facts.
Thank you for your remark. Based on it, we adjusted the conclusions See line 380-397.
Thanks again for your comments. We hope the article quality has improved after taking them into consideration.
Reviewer 2 Report
Comments and Suggestions for Authors
The manuscript under review, "Relationships between Physical Activity Knowledge and Personal Habits with Recommendations for Patients: Evidence from Primary Care Physicians", highlights the results of a cross-sectional survey of primary care physicians regarding their personal PA habits and recommendations to their patients. Overall the manuscript is well-written and successfully meets the objectives of the study.
Minor concerns:
Lines 20-22: The sentence beginning "The results..." is a little unclear for the abstract. I got lost in what related to the PCPs and what related to their patients. In addition, please change "statistically independent" (line 22) to "not statistically dependent".
Materials & Methods: There are 2 sections numbered 2.4.
Line 134: What does the "(V)" stand for?
Line 139: "Chi-squared" should be "Chi-square".
Line 140: Please explain the "provision of PA recommendations". Is this a dichotomization of recommended PA/not recommended PA?
Table 1: Please be consistent in the use of American or European decimal notation.
Table 2: The Chi-square test is not appropriate for the BMI * Categories of PA habits tests due to more than 25% of the cells have expected counts of 5 or fewer. Please use Fishers Exact and add to the statistical methods.
Table 4: The footnote for the table repeats the ** and does not have a line for the "*'.
Table 5: The means and standard deviations for the self-reported knowledge and self-reported competence are the same for the entire table. Please check the values.
Line 217: "...significantly influenced how frequently they made recommendations..." is a little misleading. If the logistic models are predicting recommend / do not recommend PA, then the sentence should be clarified.
Table 6 & Lines 214-218: Why were the independent variables dichotomized when they are all reported with all levels in every other section? Please discuss the reasoning and grouping in the statistical methods.
Line 277-278: This sentence discusses weight gain, but no data was collected on weight gain/loss/maintenance.
Lines 335-342: Please consider adding to the limitations that the level of intensity of exercise was not gathered from the PCPs.
Author Response
Dear Reviewer,
Thank you very much for taking the time to review this manuscript. You will find the answers below, and the relevant changes/corrections are included in the resubmitted version of the article.
Lines 20-22: The sentence beginning "The results..." is a little unclear for the abstract. I got lost in what related to the PCPs and what related to their patients. In addition, please change "statistically independent" (line 22) to "not statistically dependent". We have changed this part of the text in the abstract. Materials & Methods: There are 2 sections numbered 2.4. We have changed this part of the text.
Line 134: What does the "(V)" stand for? It was a translation error, we changed it to M (arithmetic mean) in the text. See line 160.
Line 139: "Chi-squared" should be "Chi-square". Thank you for your comment, we have changed it.
Line 140: Please explain the "provision of PA recommendations". Is this a dichotomization of recommended PA/not recommended PA? No, this is not dichotomization. We have made corrections in the text, see line 165-168.
Table 1: Please be consistent in the use of American or European decimal notation Thank you for your comment, we have made corrections.
Table 2: The Chi-square test is not appropriate for the BMI * Categories of PA habits tests due to more than 25% of the cells have expected counts of 5 or fewer. Please use Fishers Exact and add to the statistical methods. We have made new calculations. See table 2. We have added Fisher Exact test in 2.5, see line 163.
Table 4: The footnote for the table repeats the ** and does not have a line for the "*'. Thank you for your comment, we have made corrections. Table 5: The means and standard deviations for the self-reported knowledge and self-reported competence are the same for the entire table. Please check the values. Thank you, yes, it was an oversight on our part. We changed it. See table 5.
Line 217: "...significantly influenced how frequently they made recommendations..." is a little misleading. If the logistic models are predicting recommend / do not recommend PA, then the sentence should be clarified. Thanks for this note, the text has been corrected. See lines 240-243.
Table 6 & Lines 214-218: Why were the independent variables dichotomized when they are all reported with all levels in every other section? Please discuss the reasoning and grouping in the statistical methods. We have decided to group some variables for this test in order to achieve statistical stability, to get simplification of some results, therefore making the findings more applicable in practice. Also grouping was chosen because similar studies have grouped the variables this way.
Line 277-278: This sentence discusses weight gain, but no data was collected on weight gain/loss/maintenance We agree with your comment. We have corrected the text. See lines 307-314.
Lines 335-342: Please consider adding to the limitations that the level of intensity of exercise was not gathered from the PCPs. Thank you for noticing. We submitted it to limitations. See line 372-374.
Thanks again for your comments. We hope the article quality has improved after taking them into consideration. |
Reviewer 3 Report
Comments and Suggestions for Authors
I think the examples of the benefits and harms of physical activity and physical inactivity are redundant.
What do the authors mean by recommending physical activity?
I don't know to what extent doctors can recommend or prescribe physical activity. Doctors are not trained to do this. Only a properly accredited and trained exercise technician can do this.
The authors need to strengthen the gap in the literature that this study fills.
They also need to strengthen the study's objectives in order to convince readers to continue reading.
The sample size calculation should be introduced.
The procedures associated with the study are not sufficiently described to allow for replication.
Data was collected via an online questionnaire. The authors also asked for information on blood pressure. This would imply that the participants had a meter. Eligibility criteria are lacking.
What is the theoretical rationale for defining the correlation ranges?
Author Response
Dear Reviewer,
Thank you very much for taking the time to review this manuscript. You will find the answers below, and the relevant changes/corrections are included in the resubmitted version of the article.
I think the examples of the benefits and harms of physical activity and physical inactivity are redundant.
Thanks for this note. We fully agree with it and have limited them in the text.
What do the authors mean by recommending physical activity?
We are not sure that we understood your question correctly, but we used term ,,PA recommendations“ bearing in mind that PCPs should recommend PA as a means of improving health in addition to various treatment methods when counseling patients.
I don't know to what extent doctors can recommend or prescribe physical activity. Doctors are not trained to do this. Only a properly accredited and trained exercise technician can do this.
We cannot fully agree with you on this point – even though specialist exercise professionals are specifically qualified to design and monitor precise exercise programs, PCPs are essential in advising PA as part of comprehensive health care. When necessary, they can and should consult patients on this topic as well as encourage them to make lifestyle adjustments. This cooperative approach combines medical guidance with specialized exercise instruction to provide patients with comprehensive care. And as part of their medical education and continuous professional development, doctors in Lithuania are educated to provide advice on physical activity (PA) - it is in their curriculum.
The authors need to strengthen the gap in the literature that this study fills.
Thank you for your feedback. We filled this gap in the introductory and discussion parts.
They also need to strengthen the study's objectives in order to convince readers to continue reading.
We adjusted the aim of the study, based it more broadly on the relevant tasks. See lines 76-81.
The sample size calculation should be introduced.
Thank you for your feedback. We included it in the method section. See lines 99-101.
The procedures associated with the study are not sufficiently described to allow for replication.
Thanks for this note. We have supplemented the text by presenting the questions to be asked. See lines 137-140; 148-153.
Data was collected via an online questionnaire. The authors also asked for information on blood pressure. This would imply that the participants had a meter. Eligibility criteria are lacking.
When describing how information was collected (see subsection 2.4) on blood pressure, we indicated that we relied on self-reported information provided by the participants. Such a decision was made knowing that the subjects are people who can measure their blood pressure every day - so they can record it perfectly when filling out the questionnaire.
What is the theoretical rationale for defining the correlation ranges?
Thank you for your feedback. We have indicated the source we relied on when extracting the correlation ranges. See line 171.
Thanks again for your comments. We hope the article quality has improved after taking them into consideration.
Reviewer 4 Report
Comments and Suggestions for Authors
General Comments
I'd like to start by thanking you for the opportunity to review this manuscript.
I congratulate the authors on their work, which is very pertinent and makes it possible to give greater visibility to the still unknown importance of physical activity in the medical community.
Nevertheless, the manuscript could still be improved, namely by reinforcing the methodology in the choice and development of the questions applied (from questionnaires not previously validated), as well as deepening the discussion, which could further enrich the manuscript.
Best wishes for a job well done.
Materials and Methods
Page 2, lines 81-82 - The authors state that the representativeness of the sample was ensured, which is a huge strength of this manuscript. However, this point is not entirely clear, is the sample representative of all PCPs in the country or of that city itself?
If the intention is to represent the whole country, wouldn't it be necessary to collect data from PCPs in different parts of the country (whose social and cultural differences might impact doctors' behavior)?
How was the representativeness of the sample calculated? It's not enough to say that it's representative, the authors must explain the calculation method that allow them to be sure of this representativeness.
Page 3, lines 118-129 - The authors opted to collect data through questionnaires, partly using previously validated questionnaires and partly creating new tools.
Which 3 questions were created for the assessment of knowledge regarding PA, each with one correct answer? The presentation of these questions will allow them to be replicated in the future.
What are the 2 questions for assessing to self-report their level of competence 124 and knowledge about PA?
And, which are the 5 questions to measure the frequency of actions when counselling patients on PA issues?
In addition, the presentation of these questions will allow them to be replicated in future studies. It is equally (or even more) important to understand how they were developed, since we are working on the development (and supposed validation) of new questionnaires. What procedures did the authors carry out to develop and validate these questions/questionnaires?
Pages 3 and 4, lines 143-145 - Please add a reference to support the classification of r’s values.
Page 5, lines 175 and 182 - Please add a space between the table’s end and the next sentence. Please do the same for the following tables.
Page 6, line 199 - Add a point at the sentence’ end.
Page 7, line 219 - Remove the extra space between “patiants frequency”, and please correct the word “patients”.
Page 7, table 6 –- Please add a note identifying the “OR” and “CI”.
Page 8, lines 242-248 - What are the possible reasons why, by 2024, only around 30 per cent of doctors will consciously understand the importance of physical activity? Could this be a gap in their academic training? Could it be that these doctors have had a non-active lifestyle since childhood and adolescence, which could make them ‘minimise’ the importance of PA? These ideas are just speculations, but it would be interesting to understand the causes of this ‘unimportance’. This study does a great job of identifying the ‘wound’ of ignorance, but we have yet to discover the cause of this wound so that we can really treat it. I would ask the authors to reflect on this comment.
Page 8, lines 249-251 - This result is very interesting as it goes against the natural tendency of the general population. As we get older there is a tendency to do more PA, but in the present study older doctors are more active than younger ones. What are the possible reasons for this result?
Author Response
Dear Reviewer,
Thank you very much for taking the time to review this manuscript. You will find the answers below, and the relevant changes/corrections are included in the resubmitted version of the article.
Materials and Methods
Page 2, lines 81-82 - The authors state that the representativeness of the sample was ensured, which is a huge strength of this manuscript. However, this point is not entirely clear, is the sample representative of all PCPs in the country or of that city itself?
If the intention is to represent the whole country, wouldn't it be necessary to collect data from PCPs in different parts of the country (whose social and cultural differences might impact doctors' behavior)?
In Section 2, when introducing participants, we emphasized that the sample reflects only PCPs working in Kaunas. See lines 90-101.
How was the representativeness of the sample calculated? It's not enough to say that it's representative, the authors must explain the calculation method that allow them to be sure of this representativeness.
The text is supplemented by providing how the sample was calculated. See line 98 – 101.
Page 3, lines 118-129 - The authors opted to collect data through questionnaires, partly using previously validated questionnaires and partly creating new tools.
Which 3 questions were created for the assessment of knowledge regarding PA, each with one correct answer? The presentation of these questions will allow them to be replicated in the future.
What are the 2 questions for assessing to self-report their level of competence 124 and knowledge about PA?
And, which are the 5 questions to measure the frequency of actions when counselling patients on PA issues?
In addition, the presentation of these questions will allow them to be replicated in future studies. It is equally (or even more) important to understand how they were developed, since we are working on the development (and supposed validation) of new questionnaires. What procedures did the authors carry out to develop and validate these questions/questionnaires?
Thanks for this observation. In Section 2, subsection 2.4. Measures, we supplemented the text with questions. See lines 137-140; 146-153.
Pages 3 and 4, lines 143-145 - Please add a reference to support the classification of r’s values.
Thanks for this observation. Reference was added, see line 171.
Page 5, lines 175 and 182 - Please add a space between the table’s end and the next sentence. Please do the same for the following tables.
Thanks for this observation. Spaces are fixed.
Page 6, line 199 - Add a point at the sentence’ end.
Thanks for this observation. We corrected it.
Page 7, line 219 - Remove the extra space between “patiants frequency”, and please correct the word “patients”.
Thanks for this observation. We corrected it.
Page 7, table 6 –- Please add a note identifying the “OR” and “CI”.
Note was added. See line 245.
Page 8, lines 242-248 - What are the possible reasons why, by 2024, only around 30 per cent of doctors will consciously understand the importance of physical activity? Could this be a gap in their academic training? Could it be that these doctors have had a non-active lifestyle since childhood and adolescence, which could make them ‘minimise’ the importance of PA? These ideas are just speculations, but it would be interesting to understand the causes of this ‘unimportance’. This study does a great job of identifying the ‘wound’ of ignorance, but we have yet to discover the cause of this wound so that we can really treat it. I would ask the authors to reflect on this comment.
Thanks for this comment. We agree that this is an important issue among doctors, however our study did not focus on this topic. However, in the discussion section, see lines 274-276, we present possible reasons for the lack of PA.
Page 8, lines 249-251 - This result is very interesting as it goes against the natural tendency of the general population. As we get older there is a tendency to do more PA, but in the present study older doctors are more active than younger ones. What are the possible reasons for this result?
Thanks for this comment. We commented on these reasons in the discussion section. See lines 281-283.
Thanks again for your comments. We hope the article quality has improved after taking them into consideration.
Round 2
Reviewer 1 Report
Comments and Suggestions for Authors
Several of my review remarks were and are of a "future" nature, i.e. drawing the authors' attention to universal methodological and logical regularities. Although not all of them have been used in the corrected version of the manuscript, I think that they will be useful to the authors when writing their next works. I believe that the amendments made have changed the quality of the manuscript sufficiently for its further processing.
Author Response
Dear Reviewer,
thank you for your time and comments. At this stage, we additionally adjusted the methodological part, which includes new sources.
We will certainly take your advice into account in our future works. Thank you.

Reviewer 3 Report
Comments and Suggestions for Authors
The authors made the suggested changes.
Author Response
Dear Reviewer, Thank you for your time and comments.
Authors
Reviewer 4 Report
Comments and Suggestions for Authors
General Comments
I would like to thank the authors for their work and consideration of my comments.
The article addresses an interesting topic, its relevance is justified and the results are interesting.
However, I consider that it still presents a methodological weakness linked to the use of non-validated questionnaires. The authors must reinforce their methodology by explaining the various phases of development and validation of the questionnaires they created specifically for this study.
Best wishes.
Materials and Methods
Measures
It is equally (or even more) important to understand how they were developed, since we are working on the development (and supposed validation) of new questionnaires. What procedures did the authors carry out to develop and validate these questions/questionnaires? This question was not explained during the 1st round of reviews.
Discussion
Line 470 – Please add a comma after “i.e.”.
Author Response
Dear reviewer,
Thanks for the additional comments. While describing the structure of the questionnaire, we additionally explained the logic of questions, and additionally calculated the psychometric characteristics of the parts. See subsection 2.4. Measures. We also corrected a typographical error.
Authors.
Authors.

Round 3
Reviewer 4 Report
Comments and Suggestions for Authors
I would like to thank the authors for considering my last comments.
The methodological weakness of using questionnaires that have not been validated and that have not gone through all the steps to be validated (i.e. exploratory and confirmatory factor analysis) remains, although the authors have endeavoured to explain the steps followed and have presented the questions used clearly.
Best wishes for the continuation of your excellent work.